# Convergent evolutionary patterns of heterostyly across angiosperms support the pollination-precision hypothesis

Violeta Simón-Porcar [1,2] ✉, Marcial Escudero[1], Rocío Santos-Gally [3], Hervé Sauquet [4], Jürg Schönenberger [5], Steven D. Johnson[2] & Juan Arroyo [1] ✉

Since the insights by Charles Darwin, heterostyly, a floral polymorphism with morphs bearing stigmas and anthers at reciprocal heights, has become a model system for the study of natural selection. Based on his archetypal heterostylous flower, including regular symmetry, few stamens and a tube, Darwin hypothesised that heterostyly evolved to promote outcrossing through efficient pollen transfer between morphs involving different areas of a pollinator's body, thus proposing his seminal pollination-precision hypothesis. Here we update the number of heterostylous and other style-length polymorphic taxa to 247 genera belonging to 34 families, notably expanding known cases by 20%. Using phylogenetic and comparative analyses across the angiosperms, we show numerous independent origins of style-length polymorphism associated with actinomorphic, tubular flowers with a low number of sex organs, stamens fused to the corolla, and pollination by long-tongued insects. These associations provide support for the Darwinian pollination-precision hypothesis as a basis for convergent evolution of heterostyly across angiosperms.

The repeated nature of floral evolution is evident from the scattered phylogenetic distribution of many floral traits[1–6], of pollination modes and syndromes[3,7–10] as well as of sexual, breeding and mating systems[11–18]. A major paradigm, supported by evidence at micro- and meso-evolutionary time scales, is that such convergences have evolved in response to selection for increased pollination and reproductive efficiency in different lineages with a common selection regime[19–22]. According to the pollination-precision hypothesis, reproductive efficiency is increased by floral traits improving the fit of pollinators to flowers and their reliable contact with sex organs, thus improving pollen deposition on precise areas of pollinators' bodies[23–25]. To unravel the evolutionary significance and adaptive meaning of important floral features that have evolved repeatedly across angiosperms, it is necessary to quantify and characterise the macro-evolutionary patterns and correlates of these features.

Heterostyly is a polymorphic breeding system well established as a valuable model system in evolutionary biology since Darwin's book 'The different forms of flowers on plants of the same species'[26–29]. It consists of the existence of two (distyly) or three (tristyly) hermaphroditic floral morphs in a population, whose essential feature is a discrete differentiation in stigma and anther heights, which are reciprocally positioned (reciprocal herkogamy[30]). The typical heterostylous syndrome also encompasses a self- and intramorph incompatibility system (heteromorphic self-incompatibility, HetSI[31]) and various ancillary polymorphisms such as pollen sculpture and length of stigmatic papillae (ancillary traits[32]).

[1]Departamento de Biología Vegetal y Ecología, Facultad de Biología, Universidad de Sevilla, E-41080 Sevilla, Spain. [2]School of Life Sciences, University of KwaZulu-Natal, P Bag X01, Scottsville, Pietermaritzburg 3209, South Africa. [3]Conahcyt-Instituto de Ecología, UNAM, 04510 México, D.F., México. [4]National Herbarium of New South Wales, Royal Botanic Gardens and Domain Trust, Sydney, NSW, Australia. [5]Department of Botany and Biodiversity Research, University of Vienna, Rennweg 14, 1030 Vienna, Austria. ✉e-mail: violetasp@us.es; arroyo@us.es

Heterostyly is infrequent[33,34] but is widely distributed across the angiosperm tree, with several apparently independent origins that have made it a classic example of evolutionary convergence at all taxonomic levels[30,33–38]. Heterostylous taxa are typically characterised as presenting actinomorphic flowers with few stamens and a floral tube with nectar concealed at the base, suggesting some level of specialisation for animal-mediated pollination[26,30,33]. Based on this floral archetype, Darwin[26] and Lloyd and Webb[39] hypothesised that heterostyly evolved to promote cross-fertilisation through disassortative mating (as in dioecious plants but maintaining both male and female functions in the same flower) through efficient pollen transfer between morphs that involves different areas of a pollinator's body (Fig. 1). Hence, the basis of the Darwinian hypothesis about the evolutionary meaning of heterostyly was the pollination-precision hypothesis. Additionally, Ganders[33] suggested that the floral tube promotes precise contact between the plants' sex organs and the pollinators' body, and there is extensive literature e.g.,[26,39–42] regarding long-tongued pollinators probing for nectar as being agents of precise pollination in heterostylous flowers.

In addition to the interpretation of heterostyly in the context of pollination ecology, various alternative models have sought to understand heterostyly in the framework of heteromorphic self-incompatibility (HetSI). Over the last century, these models have focused on the genetic linkage between HetSI and style-length polymorphism as a means of selfing avoidance e.g.,[43–45]. Studies reconstructing the evolution of style-length polymorphisms in particular lineages have supported either the pollination ecology[46,47] or the selfing avoidance scenarios[38], with no particular overall consensus. Macroevolutionary testing for correlation of heterostyly with floral traits and pollination systems that enhance pollen deposition on precise areas of pollinators' bodies is an idea that was suggested by Lloyd and Webb[30] at a time when the analytical tools required were scarcely developed.

In this study, we used angiosperm-wide datasets on floral morphology and pollination system combined with phylogenetic information to test the pollination-precision hypothesis and assess macroevolutionary patterns of heterostyly in detail. We carried out a comprehensive and up-to-date review of reports of heterostyly and other style-length polymorphisms in genera across the angiosperms, and then used phylogenetic comparative methods to determine the distribution and number of gains and losses of style-length polymorphisms, and their correlation with six floral traits and six pollination systems. We show that style-length polymorphism originated repeatedly and independently across genera in lineages with flowers and pollination systems favouring precise pollen transfer. Our results

provide support for the pollination-precision hypothesis as a basis of the convergent evolution of heterostyly across angiosperms.

## Results

### Reports of style-length polymorphic genera

We found reports of style-length polymorphisms in 247 currently accepted genera belonging to 34 families (Fig. 2; Supplementary Data 1). Of these, 184 currently accepted genera from 27 families were already listed in the last review[34], and 63 currently accepted genera and seven families (namely: Asparagaceae, Carlemanniaceae, Ericaceae, Haemodoraceae, Loganiaceae, Olacaceae and Theaceae) are new reports of style-length polymorphism not related to taxonomic changes. Most of the reports of style-length polymorphic genera (50%) were based on descriptions with or without measurements of sex organs; 29% of records were based on mentions, and 21% included measurements subjected to statistical tests for significant differences between sex organ lengths. Reports of style-length polymorphism were strictly linked to (cryptic) dioecy in eight genera and to style-length dimorphism (non-reciprocal herkogamy) in five genera. We list doubtful reports of style-length polymorphism for further 27 genera and seven families (Supplementary Data 1).

### Macroevolution of style-length polymorphisms

The GBOTB angiosperms tree included 208 out of the 247 style-length polymorphic genera (84%). The best fitting model for the evolution of style-length polymorphism across the GBOTB tree was the 'all rates different' HMM model with two transition rate categories (Supplementary Table 1). The two transition rate categories and style-length polymorphism states are not equally probable in the tree. The R1 category (Fig. 3A) and style-length monomorphism, the ancestral state, (Fig. 3B) are much more frequent in the tree. Category R1 presents low transition rates between the style-length monomorphic and the style-length polymorphic states, and category R2 is associated with rapid transitions between both states (Fig. 3C). Although in R1 both directions (from/to style-length polymorphic state) were very infrequent, the inferred transition rate to style-length polymorphism was 100 times lower than the opposite transition. Within R2, the transition to style-length polymorphism was 2.7 times lower than the opposite (Fig. 3C). These transition rates resulted in a strikingly high number of monomorphic lineages accumulated through time in R1, and a slightly higher number of polymorphic lineages accumulated through time in R2 than in R1 (Fig. 4A). From this model, we inferred 152 independent gains of style-length polymorphism and 137 independent losses that generally appeared during most of angiosperms evolutionary time (Fig. 4B). The most ancient and most recent gains, respectively, were dated at 86.01 and 0.02 Myr ago, with a maximum probability at 6.6 Myr (mode estimated with the function locmodes of the R package multimode v1.5[48]). The most ancient and most recent losses, respectively, were dated 68.37 and 0.22 Myr ago, with a maximum probability at 6.8 Myr. The ages given above correspond to the ages estimated from a single simulation in stochastic mapping and do not include the inherent uncertainty of divergence time analyses.

### Correlated evolution of style-length polymorphism and other floral traits

In all cases, 'all rates different' models fitted better than 'equal rates' models for both the dependent and independent modes of evolution. For most floral traits analysed (Supplementary Data 2), a model of dependent evolution with style-length polymorphism received greater support than the corresponding model of independent evolution (Supplementary Table 2). The models of dependent and independent evolution obtained similar support only for 'Fusion of filaments' and 'Number of structural carpels'. The models of dependent evolution showed that style-length polymorphic genera presented higher rates of transition from the unfused perianth to the fused perianth state

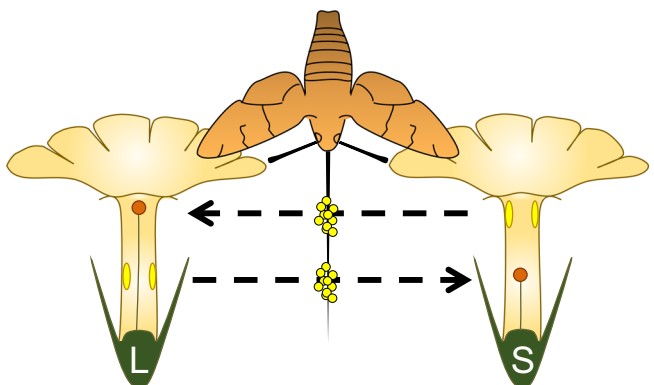

**Fig. 1 | Graphic representation of heterostyly.** Graphic representation of the two floral morphs (L=long-styled morph, S=short-styled morph) of a distylous species and the hypothetical mechanism of pollen transfer between morphs in differentiated parts of a pollinator's body, based on the pollination-precision hypothesis.

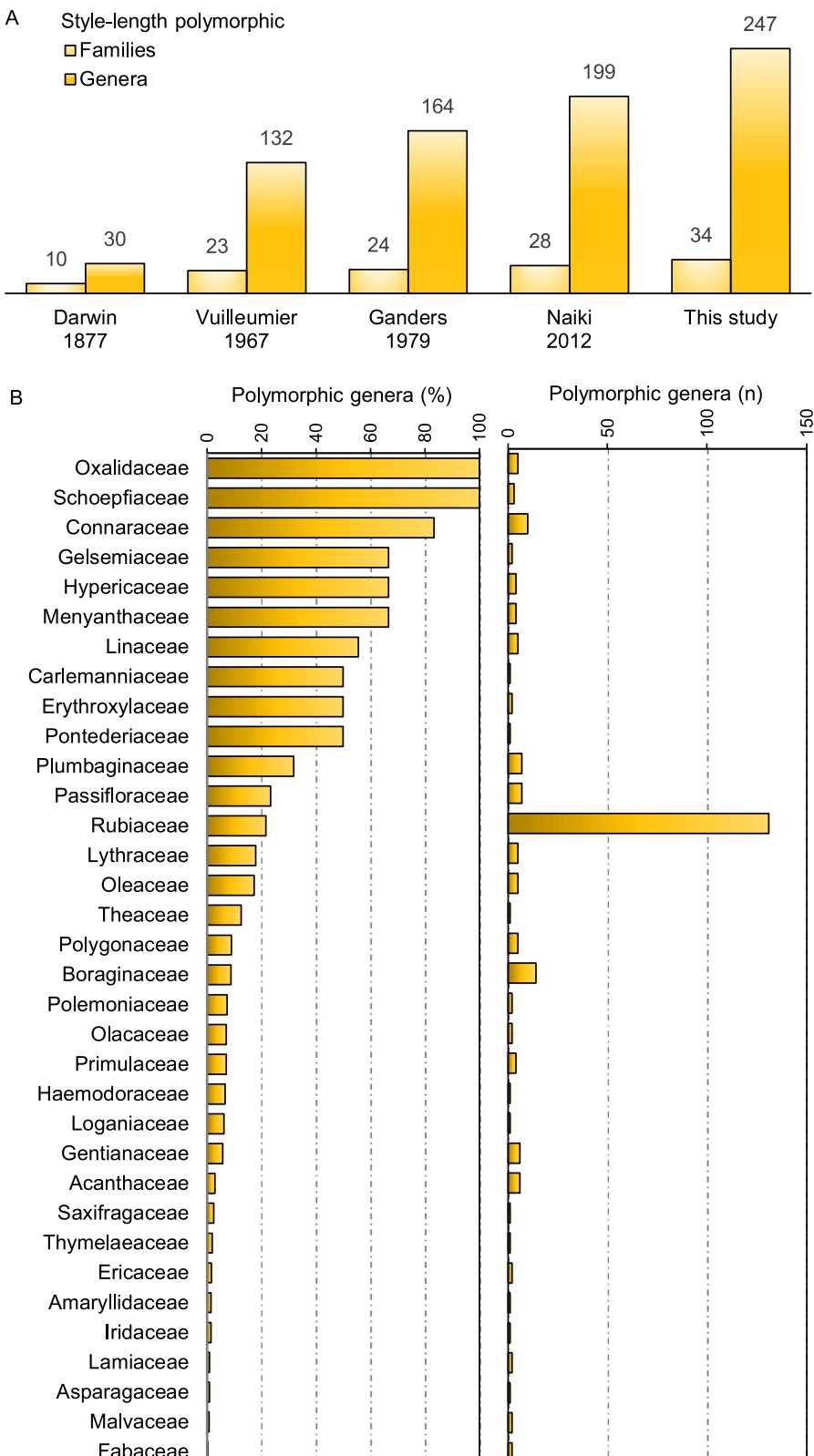

**Fig. 2 | Reports of style-length polymorphic genera.** Bar plots showing (**A**) the increasing number of genera and families containing style-length polymorphisms as reported in earlier reviews of the history of heterostyly research and in the present study; and (**B**) the percentage and the absolute number of genera containing style-length polymorphisms per family. Source data are provided as a Source Data file.

(Fig. 5A), from free stamens to stamens fused with the perianth (Fig. 5B), from zygomorphic to actinomorphic perianth (Fig. 5C), from fused to unfused stamens (Fig. 5D), from many to few stamens (Fig. 5E), and from many to few carpels (Fig. 5F) than for the opposite

transitions. The rates for the remaining transitions between states in each model also supported the highest probability for the style-length polymorphic states being associated with fused, actinomorphic perianth, few stamens with free filaments fused with the perianth, and few

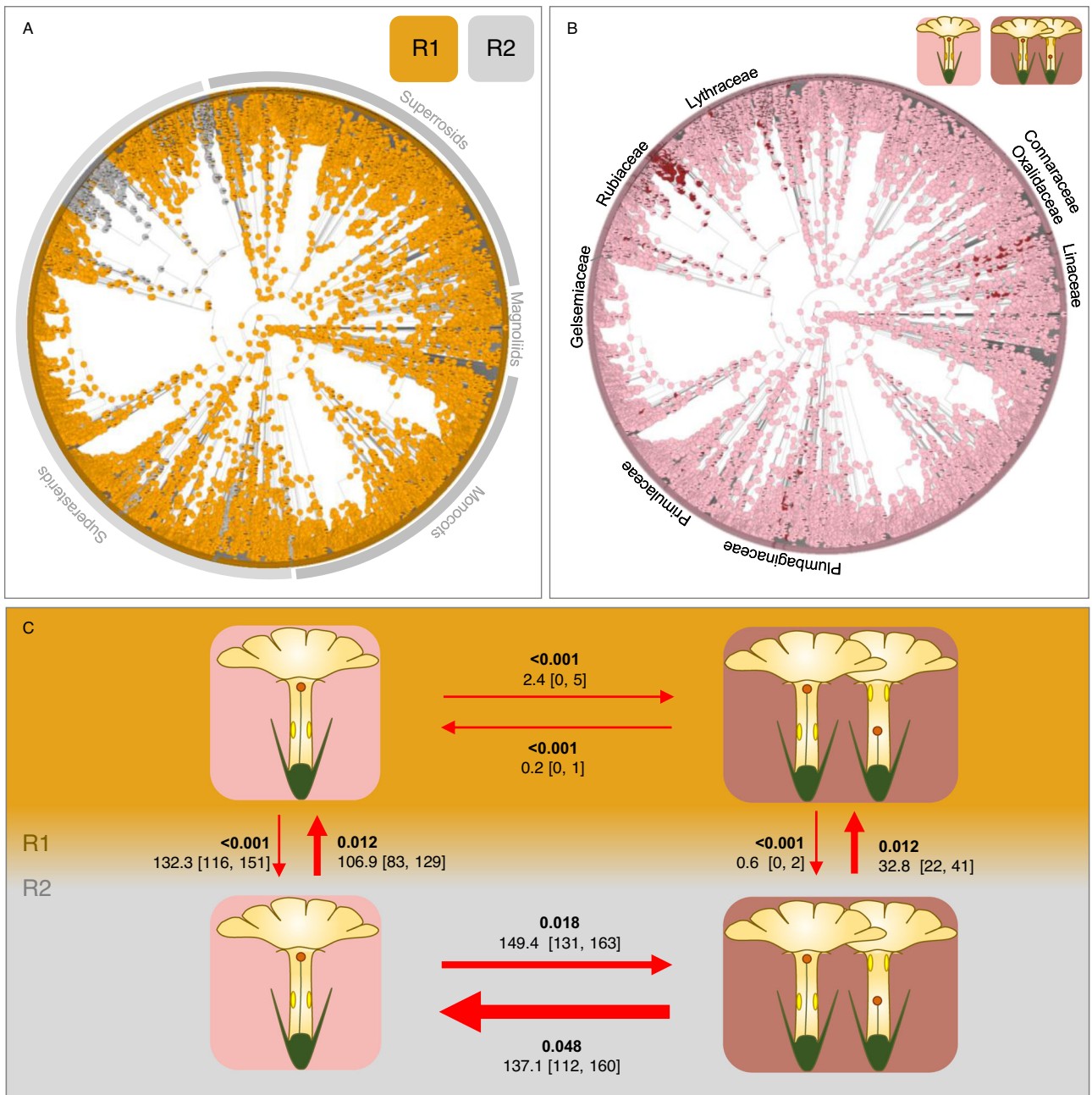

**Fig. 3 | Macroevolutionary patterns of style-length polymorphism.** Graphic representation of the 'all rates different' model fitting the evolution of style-length polymorphism across the GBOTB tree. Tree nodes are coloured as their probability of belonging to transition rates categories R1 or R2 (**A**) and the presence or absence of style-length polymorphism (**B**). Major angiosperm lineages and some style-polymorphic families have been labelled in (**A**, **B**), respectively. Transition rates between R1 and R2, and polymorphic and monomorphic states, are shown in bold in (**C**), jointly with the number of transitions and their confidence intervals based on 100 stochastic mapping simulations. Source data are provided as a Source Data file.

carpels (Fig. 5). The high transition rate from style-length polymorphism to style-length monomorphism in zygomorphic genera is especially noteworthy (Fig. 5C).

### Correlated evolution of style-length polymorphism and pollination systems

As above, 'all rates different' models fitted better than 'equal rates' models for all the dependent and independent modes of evolution. The model of dependent evolution with style-length polymorphism presented greater support than the corresponding model of independent evolution only for 'long-tongued insects' vs. 'short-tongued insects' pollination systems (Supplementary Table 3). The transition

rates between states in this model supported the highest probability of the style-length polymorphic state with long-tongued insect pollination (Fig. 6). The missing transition from style-length monomorphism to style-length polymorphism in short-tongued pollination systems is especially remarkable (Fig. 6). The models of dependent and independent evolution presented similar support for the 'biotic' vs. 'abiotic' and the 'insect' vs. 'bird' pollination systems, and in these cases the transition rates in the models of dependent evolution suggested an association of style-length polymorphism with biotic and insect pollination (Supplementary Table 3 and Supplementary Fig. 1). As expected, we found no verified report of abiotic pollination in style-length polymorphic species (Supplementary Data 3). The models of

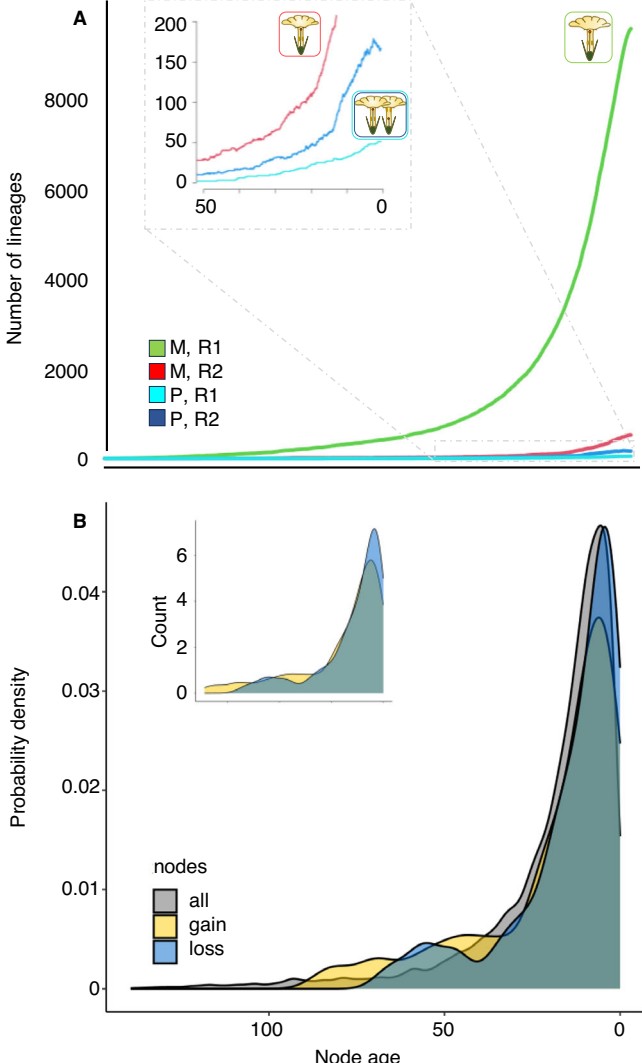

**Fig. 4 | Evolution of style-length polymorphism along the evolutionary history of angiosperms. A** Number of monomorphic (M) and style-length polymorphic (P) lineages accumulated within transition rates R1 and R2. **B** Probability density and count (inset) of the inferred gains and losses of style-length polymorphism. Source data are provided as a Source Data file.

independent evolution obtained greater support for the 'insect' vs. 'vertebrate', 'generalist' vs. 'specialist' and 'long-tongued animals' vs. 'short-tongued animals' (Supplementary Table 3).

## Discussion

The number of recognised style-length polymorphic lineages and taxa has significantly increased in the last decades, from 164 genera in 24 families in Ganders' review[33] to 247 genera in 34 families in this study. The substantial increase in records (62 accepted genera and seven families) reported in this study with respect to the former review[34] is partly due to the inclusion of cryptic dioecious (seven genera and one new family) and style-length dimorphic (five genera from five new families) taxa. Deviations from the perfect heterostylous syndrome, including the lack or partial expression of heteromorphic self-incompatibility[49–54], ancillary polymorphisms[55–57], sex organ reciprocity[46,58,59] and hermaphroditism[40,60,61] are common and have historically made it difficult to define heterostyly[29,30,62–64]. Such deviations are difficult to assess and have mostly received none or uneven attention across taxa. Often, data are available from single populations, which may not be representative of the entire species[59,65–67], and

even less so of the entire genus, the taxonomic level analysed in this study. Relative to the potential number of style-length polymorphic species, the scarcity of functional and fine morphological studies hinders a complete picture of the occurrence and distribution of these phenomena. Numerous reports of heterostyly are just incidental mentions or descriptions without available supporting data, and there are several cases in which style-length dimorphism has been referred to as heterostyly e.g.,[68] or in which their distinction is not clear e.g.,[69,70]. Hence, here we advocate for the use of style-length polymorphism as a more comprehensive term that encompasses all taxa with the most essential feature of the syndrome: a discrete variation in style-length and herkogamy[30], which is expected to promote disassortative mating[71–73].

We report 152 independent gains of style-length polymorphism across the angiosperms tree at the genus level. This number is robust for purposed of this analysis but is likely to be still an underestimation. Nonetheless, the number of estimated independent gains of style-length polymorphism increases in parallel with the number of known style-length polymorphic taxa and a better understanding of angiosperm phylogenetics. In the past, exploring the distribution of heterostyly across angiosperms with, at that time, limited phylogenetic information, Lloyd and Webb[30] estimated 23 independent origins at the family level. Later, Naiki[34] estimated between seven and 13 independent origins using an order-level angiosperm phylogeny. Moreover, molecular phylogenies have revealed cases of repeated, independent evolution of heterostyly within some families[36,38] and even lower ranked taxa[35,46,74–76]. Yet, the scarcity of phylogenetic studies at this level precludes any possible generalisation about evolutionary patterns of style-length polymorphism at the tips of the angiosperm phylogenetic tree. The high number of independent gains relative to its low frequency (ca. 2% of angiosperm genera, vs. 7% for dioecy[77]) and accumulated biological and ecological knowledge make style-length polymorphism a unique study case for understanding the evolutionary origins of homoplastic breeding systems. The multiplicity of ontogenetic patterns[78,79] and sporophytic and gametophytic self-incompatibility systems or their lack[49,51,80–82] associated with style-length polymorphism suggest diverse evolutionary mechanisms underlying its origin. As a model system for the study of supergenes, advances in new genomic data sets and comparative analyses e.g.,[83–86] are shedding light onto the molecular pathways of convergent evolution in independent style-length polymorphic lineages, and our results can help to optimise the choice of future study systems.

The inferred gains and losses of style-length polymorphism were aged along most evolutionary time in the angiosperm phylogeny (Fig. 4B). Our reconstruction was based on an HMM model with two transition rate categories from and to the style-length polymorphic state. Within rate category R1, which is dominant in the angiosperm tree, the rates of polymorphism gains and losses were very low (although the loss rate was ca. 100 times higher than the gain rate), resulting in two gains of polymorphism and no losses inferred. Within the rate category R2, which is less frequent and mostly found in shallow parts of the phylogeny, rates of polymorphism gains and losses were higher than in R1 and the loss rate was ca. 3 times higher than the gain rate, resulting in 152 gains and 137 losses inferred in R2. Overall, the numbers of gains and losses inferred are congruent with the transition rates in a context of prevalence of the ancestral monomorphic condition. Whether style-length polymorphism was associated with different speciation rates or extinction rates in R1 and R2 cannot be known with our current knowledge of phylogenetics and distribution of style-length polymorphism across all angiosperm species[87]. At the genus level there is mixed phylogenetic evidence about the role of heterostyly in diversification rates[76,88,89], see also ref. 90, which is congruent with the apparent context and/or lineage-dependent role of floral traits on angiosperms diversification rates[91].

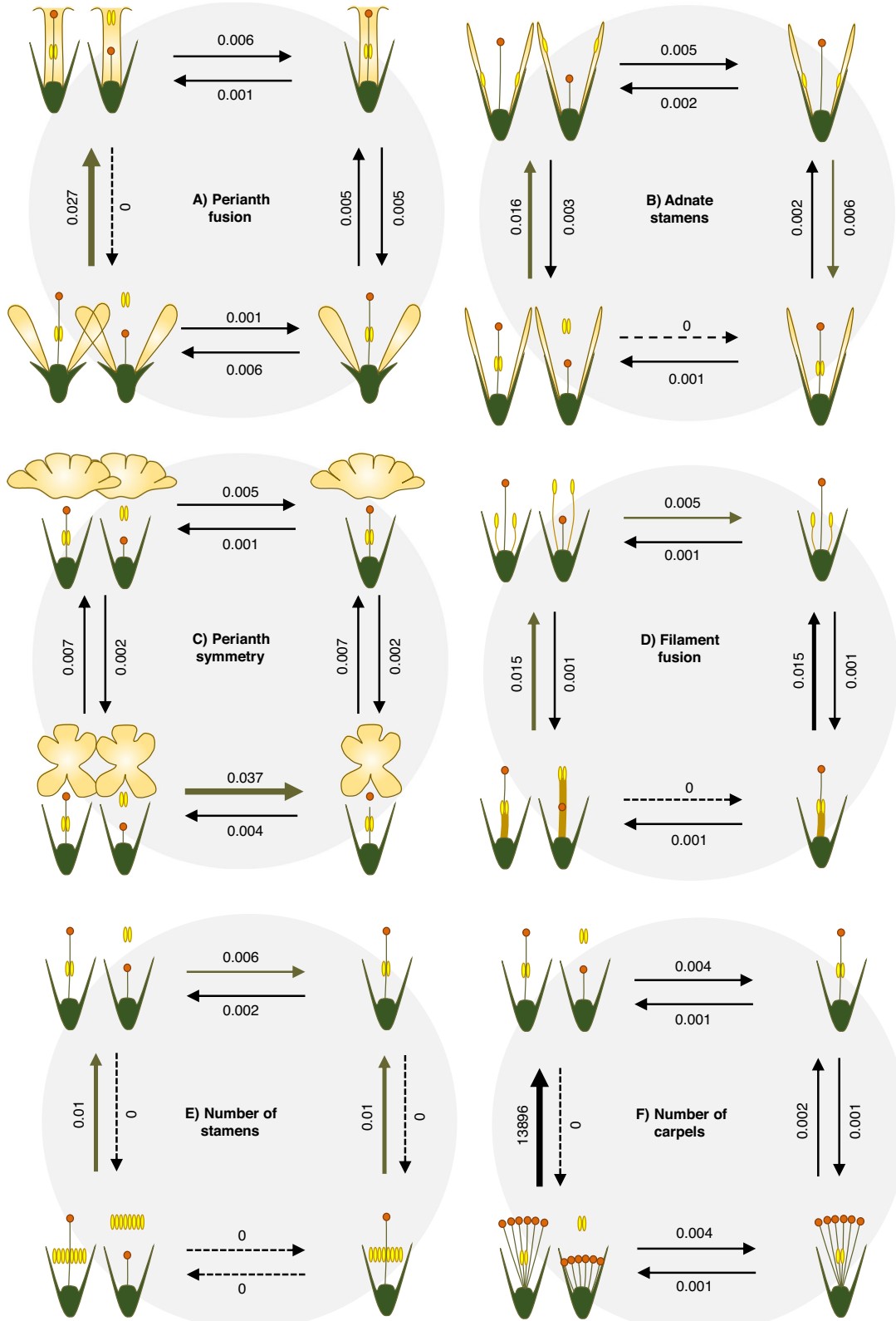

**Fig. 5 | Correlated evolution of style-length polymorphism and other floral traits.** Graphic representation of Pagel's models of correlated dependent evolution between presence of style-length polymorphism (represented through two morphs state) and 'Fusion of perianth' (**A**), 'Fusion of filaments to inner perianth series' (**B**), 'Symmetry of perianth' (**C**), 'Fusion of filaments' (**D**), 'Number of fertile stamens' (**E**) and 'Number of structural carpels' (**F**) across angiosperms. For each floral trait model, the state with the highest association with style-length polymorphism is on the top left.

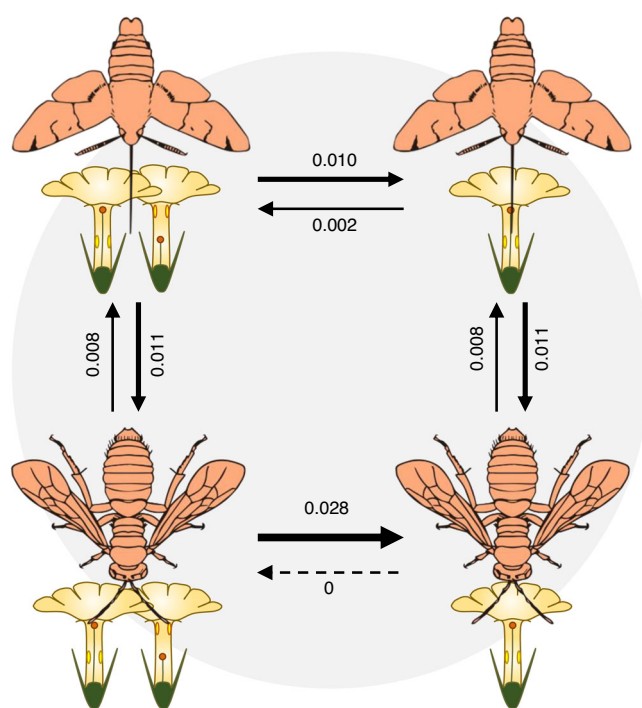

**Fig. 6 | Correlated evolution of style-length polymorphism and long-tongued insects pollination system.** Graphic representation of Pagel's model of dependent evolution of style-length polymorphism (represented through two morphs state) on 'long-tongued insect' (represented by a moth) vs. 'short-tongued insect' (represented by a bee) pollination systems. This model presented greater support than the corresponding model of independent evolution. The state with the highest association with style-length polymorphism is on the top left. Insect icons obtained from www.divulgare.net.

We found that style-length polymorphism is evolutionary correlated with various floral traits related to pollination precision and intermediate levels of specialisation for animal pollination, notably actinomorphic flowers with fused perianth parts (a proxy for a floral tube), few stamens fused to the perianth, and few carpels, and with a long-tongued insect pollination system. These results support empirically, at the macroevolutionary level, the long-standing pollination-precision hypothesis set up before the phylogenetic era by Darwin[26], Ganders[33] and Lloyd and Webb[30,39]. These authors recognised the non-random distribution of heterostyly among plant families and its prevalence in animal-pollinated species with small, actinomorphic flowers with few stamens, and presenting nectar concealed at the base of a floral tube. These apparent associations informed their hypotheses for the function of heterostyly as a means of promoting pollination efficiency, enhanced by floral tubes promoting the precise contact between pollinators probing for nectar and floral sex organs. After much microevolutionary evidence e.g.,[42,72,73,92] see also ref. 47, the broad-scale significance of these patterns has been tested here at a macroevolutionary level encompassing all major angiosperm clades. Based on solid current phylogenetic knowledge and extensive floral trait and pollination system databases, our analyses support pollination precision as the selective force behind the repeated and independent evolution of style-length polymorphisms in angiosperms.

The macroevolutionary association of style-length polymorphism and long-tongued insect pollination supports the hypothesised role of these animals as the main agents of the precise pollen transfer in heterostylous lineages. However, observational e.g.,[93] and experimental e.g.,[73] evidence suggests that short-tongued pollinators can also achieve precise pollination in certain style-length polymorphic plants. Some style-length polymorphic taxa also exhibit a diverse array of long- and short-tongued pollinators that may play complementary

roles in the pollination of each morph e.g.,[73,94,95], or whose pollination efficiency may depend on the particular flower architecture of each style-length polymorphic lineage. Future efforts assessing the macroevolutionary correlation of style-length polymorphism with pollinator types, including the assessment of their behaviour and precise pollen deposition areas e.g.,[25], their efficiency in promoting disassortative pollination as done for some specific cases e.g.,[42,96–98], and their interplay with floral traits, would seem the most promising additional test of the pollination-precision hypothesis in this context.

Besides pollination efficiency, future efforts could also be directed to assess the macroevolutionary correlations of style-length polymorphism with other plant reproductive systems[99], specifically with self-incompatibility to unravel the complementary role of selfing avoidance in the evolution of style-length polymorphisms[44,45] and dioecy to unravel the role of transitions towards dioecy in the loss of style-length polymorphism across the angiosperms[40,77]. Nearly 150 years after "The different forms of flowers", style-length polymorphisms still have much to offer as a model study system for angiosperm evolution.

## Methods
### Style-length polymorphism database
Current knowledge does not allow for reliable estimates of the occurrence of style-length polymorphisms at the species level. Thus, we focused our search at the genus level, and used the reviews of Ganders[33] and Naiki[34] to generate a preliminary list of all genera reported to contain at least one heterostylous species (hereafter style-length polymorphic genera). We updated this list in November 2020 through a systematic literature search in Google Scholar to identify further genera with style-length polymorphisms. First, we carried out a search of publications since 2012 using "heterostyly" OR "distyly" OR "tristyly" OR "heterodistyly" OR "distylous" OR "tristylous" OR "heterodistylous" OR "reciprocal herkogamy" OR "style dimorphism" OR "style polymorphism" OR "stylar dimorphism" OR "stylar polymorphism" OR "stylar morph" OR "cryptic dioecy" OR "functional dioecy" as search terms putatively related with heterostyly in the full text and examined the first 1000 most relevant entries. This search retrieved some style-length polymorphic genera that were not included in the survey by Naiki[34], and thus we carried out a second search without publishing year limitations. In this search, updated in January 2022, we used the same search terms as before plus OR "disassortative mating" OR "style-length" OR "stigma height" OR "herkogamous" as search terms in the article title, and examined the ca. 4500 retrieved entries and the references therein.

We recorded as style-length polymorphic the genera with at least one report of heterostyly or an obvious discrete style-length dimorphism and recorded whether the report was based on a mention, on a qualitative description, on a quantitative description or on morphological data subjected to statistical tests, and whether it included illustrations (Supplementary Data 1). Although the typical heterostylous syndrome includes reciprocal herkogamy, HetSI and ancillary traits, we followed a strictly morphological criterion because reciprocal herkogamy is the only requirement for the pollination-precision hypothesis. Hence, we listed genera as style-length polymorphic regardless of the presence or absence of the two latter features, or the lack of information about them. Indeed, Lloyd and Webb[30] considered a discrete variation in stigma height and a different "sequence of heights at which the anthers and stigmas are presented within their flowers" as the only defining trait of heterostyly. This definition included style-length dimorphism and we also included reports of this condition because Lloyd and Webb's model[39] see also ref. 45,82 proposes it as an evolutionary precursor of heterostyly that promotes disassortative mating in a very similar manner[71–73]. In addition, we found several cases in which style-length dimorphism had been labelled as heterostyly e.g.,[68] or instances where the distinction

between heterostyly and style-length dimorphism remained unclear even in light of available data e.g.,[69,70]. Therefore, as numerous reports were based on mentions of heterostyly with no empirical or graphical support (see Results), excluding reports of style-length dimorphism was deemed inappropriate. Following our morphological criterion, we also included some genera with style-length polymorphisms associated with cryptic and non-cryptic dioecy e.g.,[100], which could represent the evolutionary pathway from heterostyly to dioecy[40]. We only included genera with apparently fully functional sex organs and obvious style-length polymorphisms and excluded cases in which staminate individuals presented ovary reductions. Genera for which we only found explicit reports of style-length dimorphism or heterostyly associated with dioecy were coded accordingly (Supplementary Data 1). We also found some genera with doubtful information, which were listed as doubtful (Supplementary Data 1) but not included in the analyses. Genera with incorrect reports of "heterostyly" referring to continuous, intra-individual variation or among-population variation of style-length were not listed e.g.,[101,102]. The genera in the final list were searched for their latest taxonomic treatment at the Plants of the World database (powo.science.kew.org).

### Floral traits database

To compile a data set of floral traits related to pollination precision across all angiosperm families, we used the PROTEUS database of floral morphological traits, assembled within the eFLOWER initiative[103]. We used the eFLOWER dataset published by ref. 104, which included data for 29 primary floral characters in 792 species from 776 genera, 372 families (86%) and 63 out of 64 angiosperm orders recognised in APG IV[105]. For the present study, we selected ten of these primary floral traits that were related to pollination precision and expanded the data set by scoring them for an additional 231 species belonging to different style-length polymorphic genera (see ref. 104 for details on scoring methodology). Our final data set, provided as Supplementary Data 2, comprises a total of 8200 data records for 1023 species from 977 different genera (201 style-length polymorphic genera). Four traits ('Number of ovules per functional carpel', 'Number of androecium structural whorls', 'Anther orientation', and 'Style differentiation') involved too few available records for style-length polymorphic genera and were thus excluded at this stage. The remaining six primary floral traits ('Fusion of perianth', 'Symmetry of perianth', 'Number of fertile stamens', 'Fusion of filaments', 'Fusion of filaments to inner perianth series' and 'Number of structural carpels') were transformed into binary secondary traits for analyses.

We used 'Fusion of perianth' as a proxy for floral tube[106,107], and coded taxa as perianth unfused (state 0) or fused (state 1) with a threshold of 5% of petal or sepal length. Following Darwin's arguments[26], heterostyly has been traditionally associated with narrow floral tubes restricting pollinator movements and thus favouring the precise contact between flower sex organs and pollinators[33]. 'Symmetry of perianth' was scored as actinomorphic (state 0) or zygomorphic (state 1). While zygomorphic corollas are considered as highly specialised for animal pollination and also a means for increasing pollination precision[108,109], heterostyly has been associated with actinomorphic flowers, which do not constrain the access of animals from any direction and are therefore considered as presenting intermediate levels of specialisation[26,30,33]. The 'Number of fertile stamens' has been also associated with heterostyly[30] and was scored as lower (state 0) or greater (state 1) than 10. A low number of stamens has been linked to increased pollination precision, since flowers with numerous stamens could more easily contact different areas of a pollinator's body[110]. In addition, a low number of floral parts enhances the potential for floral integration, possibly facilitating the evolution of heterostyly[110–112]. A low number of stamens could also be indicative of low pollen/ovule ratios, a major proxy for outcrossing breeding system and pollination

precision[113,114]. The three remaining traits had not yet been associated with heterostyly. We hypothesise that both 'Fusion of filaments' and 'Fusion of filaments to inner perianth series' may restrict anthers oscillation during pollinator visits, thus enhancing floral integration and favouring the precise contact of anthers with pollinator bodies[111]. Both traits were coded as unfused (state 0) or fused (state 1) with a threshold of 5% of filament length. Finally, the 'Number of structural carpels' was scored as lower as (state 0) or greater (state1) than five. As increasing floral integration[111], a low number of carpels could also be indicative of pollination precision. We repeated the analyses on all floral traits (except 'Perianth symmetry') using different thresholds for binary scoring and obtained similar results. Overall, we test the hypothesis that the presence of style-length polymorphism is more likely in flowers with actinomorphic perianths, with fused perianth parts and stamens, and few sex organs.

### Pollination system database

We searched for information on the pollination systems of style-length polymorphic species. We reviewed all the references used in our former search and performed additional systematic literature searches in Google Scholar for all accepted genera in our list and for the synonyms reported as polymorphic. The basic search included "name of genus" AND "heterostyly" OR "visit" OR "pollinat*" OR "insect" OR "bird" OR "animal" OR "wind" OR "reproductive" as search terms in the full text. In certain cases, in which this search provided mostly irrelevant references (e.g., the name of the genus coincided with an animal genus or with an author surname, or the genus has been subject of intense medicinal research), we did additional searches including the family name and excluding particular keywords related to the irrelevant results retrieved. We examined the first 20 most relevant entries for each search, checked the references within relevant articles and recorded all relevant information on the pollination systems found in style-length polymorphic species. We disregarded information on monomorphic species within style-length polymorphic genera and excluded speculative accounts based on pollination syndromes, as well as records of pollination by managed insects (e.g., *Apis mellifera* and *Bombus terrestris*) or instances of pollination outside a species natural range.

Next, we searched for information on the pollination system of monomorphic species across all angiosperm families and ecosystems. We used the dataset from ref. 115 and a search in Google Scholar, using the search term "pollination system", to find original research and review articles on the pollination system of angiosperm species. We also disregarded information based on pollination syndromes, managed insects and non-native populations. Our final database included information on the pollination system of 5038 angiosperm species (196 polymorphic and 4842 monomorphic). We scored the pollination system of each species to the most detailed level possible. This information was subsequently used to code six binary pollination systems: 'biotic' vs. 'abiotic', 'insect' vs. 'vertebrate', 'insect' vs. 'bird', 'generalist' vs. 'specialist', 'long-tongued animals' vs. 'short-tongued animals' and 'long-tongued insects' vs. 'short-tongued insects'. Birds, long-tongued bees (families Apidae and Megachilidae), long-tongued flies (families Bombyliidae and Nemestridae), butterflies and moths were scored as long-tongued, and all remaining animals were scored as short-tongued. We use tongue as a generic term referring to a prominent buccal apparatus, including insect proboscis and bird beaks. When scoring from primary data, we considered generalist pollination systems those in which no particular pollinator functional type accounted for more than 66% of visits (or efficient visits, if pollination efficiency was evaluated). We tested the hypothesis that style-length polymorphism evolved in association with long-tongued insect or bird pollination systems, as these animals, probing for floral nectar, have been regarded as the main agents promoting precise pollination[26,39–42].

## Angiosperm phylogeny and evolution of style-length polymorphisms

To obtain a global angiosperm phylogeny at the genus level, we used the species-level phylogenetic tree for seed plants published by Smith and Brown[116] ('GBOTB tree'), which includes nearly 10,000 genera. We used the drop.tip function of the R package ape v5.6[117] to trim the GBOTB tree by removing all gymnosperms and randomly selecting one species per remaining angiosperm genera. The resulting tree included 208 style-length polymorphic genera (84%) listed in our review. The genera included were evenly distributed along the phylogeny and therefore no bias is expected in our main conclusions.

We estimated the transition rates between states and ancestral state of style-length polymorphism (coded as a presence/absence binary trait) across the angiosperm phylogeny with Hidden Markov models (HMM) as implemented in the function corHMM of the R package corHMM v2.8[118]. HMM models allow for some level of heterogeneity across lineages while keeping the number of parameters low. We used these models because the distribution of style-length polymorphism across angiosperm phylogeny suggested high heterogeneity in the evolution of the trait. We ran the 'equal rates' (equivalent to the 'symmetric rates' in the case of binary traits) and 'all rates different' models with a single or two transition rate categories each (four models in total). We used the Akaike information criterion (AIC) to select the best fitting model, which was used to reconstruct the evolution of style-length polymorphism in the phylogeny through stochastic character mapping. We run 100 simulations to estimate the number and age of gains and losses of style-length polymorphism across angiosperm genera using the makeSimmap function in corHMM. We plotted the number of lineages through time within each state and transition rate with the function ltt.plot in ape, and the probability density and count of the inferred gains and losses with the function ggplot of the R package ggplot2 v3.4.1[119].

## Evolutionary correlation of style-length polymorphism with floral traits

We tested the correlated evolution of style-length polymorphism with each of the six floral traits recorded at the genus level. We accordingly assigned the floral traits recorded for each species in our dataset with the corresponding genus in our phylogeny and, in the few cases where the floral traits dataset included more than one species per genus, we selected one species randomly. We used Pagel's models[120] for two binary traits as implemented in the function fitPagel in the R package phytools v1.5.1[121]. We used fitPagel instead of corHMM because (i) they are equivalent when using a single transition rate category, and (ii) corHMM models with two transition rate categories included too many parameters and initial trials displayed a low fit to our data. We ran eight different models to estimate transitions rates among character states under alternative scenarios of independent (two models: 'equal rates' and 'all rates different') and dependent evolution (six models: the three models "x depend on y and vice versa", "x depend on y" and "y depend on x", each for 'equal rates' and for 'all rates different' options) of style-length polymorphism and the corresponding floral trait, and selected the best fitting model based on their AIC value.

## Evolutionary correlation of style-length polymorphism with pollination systems

We tested the correlated evolution of style-length polymorphism with each of the six contrasting pollination system pairs scored at the species level. For this aim, we built species-level phylogenies for each contrasting pollination systems pair analysed by trimming the original GBOTB phylogeny with the drop.tip function in ape. From our pollination system database, 1495 species were included in the GBOTB tree (Supplementary Data 3). We ran Pagel's models to estimate transitions rates among character states under alternative scenarios as explained above.

## Reporting summary

Further information on research design is available in the Nature Portfolio Reporting Summary linked to this article.

## Data availability

All data supporting the findings of this study are available within the paper and its Supplementary Information. Source data are provided with this paper.

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

## Acknowledgements

This project has received funding from the European Union's Horizon 2020 research and innovation programme under grant agreement No 897890 FLAXMaTE (V.S.-P., J.A. and S.D.J.). This work was also funded by MICINN-FEDER grants PID2021-122715NB-I00 DiversiChrom, PGC2018 099608 B 100 "REPROGRAD", CGL2013-45037-P "CONFLISEX" (M.E. and J.A.) and "Juan de la Cierva" grant IJC2018-037903-I (V.S.-P. and J.A.). Lawrence D. Harder provided advice in the scoring of pollination systems.

## Author contributions

J.A. conceived the idea; V.S.-P. performed the systematic review, recorded data on floral traits and pollination systems for heterostylous taxa and compiled the database of pollination systems across all angiosperms, with contributions of R.S.-G., J.A. and S.D.J.; H.S. and J.S. provided access to the PROTEUS database and compiled the dataset of floral traits; M.E. analysed the data with contributions of V.S.-P.; V.S.-P. produced the figures and wrote the first version of the manuscript. All authors revised, contributed to and accepted the final version of the manuscript.

## Competing interests

The authors declare no competing interests.
