## [Peer Review File · Nature Communications]

Convergent evolutionary patterns of heterostyly across angiosperms support the pollination-precision hypothesisReviewers' Comments:

Reviewer #1:

Remarks to the Author:

General comments

The submitted manuscript by Simon-Porcar et al. investigates the macroevolutionary patterns of heterostyly, specifically genera that polymorphic for style length, across the angiosperms. The authors performed an extensive literature survey, multiple times, to find new cases of heterostyly, either documented with data or mentioned in studies. The known cases were expanded by about 1/5th. Different phylogenetic comparative methods were performed and multiple gains and losses were observed, more than previously shown/suggested. Analyses also showed gains associated with other floral traits.

Overall, I first want to applaud the authors for the level of detail in searching the literature to make sure that all known cases were included in the study. With such a large-scale investigation, obtaining the necessary amount of data is no trivial task. I personally think the Smith and Brown 2018 phylogeny is the best large phylogeny to use for most inferences. However, with only 85% of the polymorphic genera included, I have to ask if using the Janssens et al 2020 "A large-scale species level dated angiosperm phylogeny for evolutionary and ecological analyses" would be any better in terms of inclusiveness. I am not aware of as many taxonomic issues in the Smith and Brown phylogeny as was in the Zanne et al phylogeny that preceded it in large-scale analyses, but could a lack of taxonomic certainty in revised clades cause a drop in some retention of targeted genera? Secondly, while I agree that providing a species level investigation is impossible, I am left wondering if the decision to prune to one tip per genus is artificially pushing more transition events to the roughly 4 million year mark? Based on the explained methodology, transitions in any given genus, regardless of size, age, and actual number of heterostyly transitions, would only be evident at the tip; regardless of the true evolutionary history of the trait or the possibility of multiple transitions occurring. For large, old genera I could see this really impacting the inferred dates. Using a large consensus tree such as the Smith and Brown tree would likely not be suitable for investigating the number of transitions within a genus due to low support values in some cases.

Another major point I want to bring up is the difficulty in translating a "continuous" trait (stamen and carpel number) to a binary trait. My primary concern is if there is a biological threshold that makes a total of 10 stamens (and similar to five carpels) important. I understand the argument about lower number of floral parts, however is there any biological difference between 9 and 10 stamens or 10 and 11 stamens? Reconstructing stamen numbers as a "continuous" trait could be relevant, and having a better understanding the model of change (Brownian vs OU vs Early Burst) might tell something.

Lastly, I do want to acknowledge the impressive amount of work that went into the manuscript, especially the data collection. However, after reading the manuscript I felt myself wanting more in terms of analyses and correlations/statements could really be made. While there is nothing wrong with the reconstruction methods, the title and summary suggest evidence for the association of heterostyly with specific animal pollination principles. This would be a fantastic thing to show, but on line 445 the authors even say that their results are independent of animal pollinators related to style-length polymorphisms. While the data is certainly useful on a large scale, and the authors suggest future studies that would be possible, I'm not sure the expectations of the title were met.

Specific comments

Line 51-53: The last sentence of the summary talks about the association between trait evolution and animal pollination, yet by the end of the manuscript these two things are clearly not compared at all. While the hypothesis is attributed to Darwin, it does seem a little bit misleading since this cannot/was not addressed with the data on hand.

Line 100: Seems a little bit odd to have just two citations that are 115 years apart. Are these the only two hypothesized examples?

Line 121: Unprecedented detail or unprecedented level/scope?

Line 158: There are several complex sentences here, so I'm a bit confused about what "it" is referring to here.

Line 179: The website appears to be powo.science.kew.org

Line 244-246: Maybe I'm missing something but why was corHHM used in some cases, while ML (ASR?) and stochastic mapping were used in others? Also, no details were provided as to what programs were used for these later analyses. I'm guessing phytools.

Line 249-250: Did the age estimates include the uncertainty in divergence times from the Smith and Brown tree, or what is just a point estimate for the consecutive nodes?

Line 256: Truly random? If more than three, why not take the majority consensus?

Line 258: version of phytools?

Line 307: Is this maximum probability value appropriate since the events are not entirely independent?

Line 312: This partially answers a previous question in regards to confidence intervals. How much confidence can we have then in these point estimates?

Line 313 (Figure 3): If the tree in one panel is going to have clade arc labels, it would be helpful if both had major lineages/important lineages marked.

Line 385: Agreed, especially if within a genus there were multiple independent transitions, which large genera may likely be the case.

Line 407: I know acctran and deltran are parsimony based, but could this observation be due to a bias in the inference method, somehow tied to the speed of evolution? Or is it due to the pruning method? If more than one species per genus was kept, its reasonable for changes involving multiple species to occur not at tips but a few nodes back.

Line 415-416: This is a bit confusing since this sentence suggest gains and losses are almost equal, but then the next sentence is 102:25 ratio.

Line 418: I'm not sure if I buy this argument entirely. Without a reliable fossil representation, this seems a bit oversimplified given any transition in a genus would only be evident at the tip, regardless of the size and age of a given genus and where in its evolutionary history one or multiple transitions occurred.

Line 420-422: While fundamentally I agree with this sentence, providing some citations or specific reasoning would be useful.

Line 446: knowing the specific selective pressures is difficult. If it is floral architecture, there is likely some selective forces that are driving the change. Is it outside the realm of feasibility to look at style-length polymorphisms vs selfing/outcrossing or more importantly animal pollination versus other vectors?

Line 454-455: This partly addresses my previous question, even if this sort of question/association is

highly interesting.

Reviewer #2:

Remarks to the Author:

The paper provides new, valuable results on the distribution of heterostyly in angiosperms as well as the number and timing of origins and losses of the trait.

The paper is well written and clear.

The analyses appear to be thorough and well executed.

In general, the paper represents a valuable contribution to studies of floral heteromorphism.

However, I wonder whether the main conclusion of the paper, namely, that heterostyly evolved in response to selection for increased pollination precision, is warranted. I will try to unpack the sources of my concern.

On the concept of "precision of pollination". The authors state that Darwin proposed the hypothesis of increased pollination precision for the origin of heterostyly. However, did Darwin really focus on "pollination precision" or, rather, disassortative pollination and promotion of outcrossing, which decrease the harmful effects of inbreeding? The hypothesis of disassortative pollination (hence promotion of outcrossing) is a quantitative model requiring that more pollen grains are transferred to reciprocal and compatible stigmas than to non-reciprocal and incompatible stigmas.

I think that a general problem affecting the main conclusion of the paper is that the authors fail to define what they mean with "pollination precision", and this omission needs to be corrected. Importantly, the authors should link their hypothesis with other studies that have addressed the role of pollination precision in angiosperm evolution and the importance of precision of sexual organ reciprocity in heterostylous taxa (for example, Armbruster et al., *New Phytologist*, 2022 and 2017, respectively).

In different parts of the paper the authors refer to "floral tubes promoting the precise contact between pollinators and floral sexual organs" (line 438), but again they do not define what they mean with "precise contact". If they mean to say that pollen is deposited on discrete, non-overlapping parts of the pollinators' body, as represented in their Fig. 1, this is not the case, or, rather, it is not as simple as that (see quantitative definition of disassortative pollination above). Some of the few studies that quantified disassortative pollination via pollen counts found that, while the hypothesis of disassortative pollination generally holds, pollen grains are far from being deposited in discrete, non-overlapping parts of the pollinators' bodies. Indeed, pollen grains from high and low anthers are found in the same parts of the pollinators' bodies, but in different proportions (see Keller et al. 2014, *Functional Ecology*).

Additionally, I wonder how one can reconcile the concept of "pollination precision" as the major driver for the evolution of heterostyly (rather than disassortative pollination and outcrossing) with the fact that, as the authors acknowledge (line 453-454), heterostylous taxa are typically pollinated by multiple pollinators and that pollen deposition on their bodies does not occur in discrete pollen packages (as for example, pollinia), but rather in a rather messy "pollen cloud" (as discussed above).

Moreover, if pollination precision were the key driver for the repeated origin of heterostyly, it would imply that heterostyly might also spur an increase of speciation rates via a form of mechanical reproductive isolation linked with the "precise" partitioning of flower-pollinator space for the uptake and deposition of pollen grains. However, previous studies found complex results on whether the

precision of pollen transfer can drive divergence. A modeling study concluded that precision of pollen transfer can have a small effect on population divergence, but this effect is very context-dependent (Haller et al., 2014, PLOS One). A macroevolutionary study found that heterostyly is linked with an increase of diversification rates, but this effect does not depend on an increase of speciation rates, rather on a decrease of extinction rates, possibly because heterostylous taxa are more resilient to changes of environmental conditions because they harbor more genetic diversity than non-heterostylous taxa (deVos et al., 2014. Proc. R. Soc. B).

The main thesis of the article is that heterostyly evolved in response to selection for increased pollination precision (rather than to promote disassortative pollination and prevent selfing, with the combined effect of increased outcrossing). If indeed pollination precision is the driver for the evolution of heterostyly, then the authors should also discuss the specific selective advantages for heterostylous species (as compared to non-heterostylous species) of having increased "precision" vs. the advantages of elevated outcrossing, which does not directly depend on "precision" per se.

To conclude, I think the authors need to provide a precise definition of what they mean with "pollination precision" and how they distinguish its selective advantages from those of disassortative pollination and outcrossing. They should also distinguish the concepts of pollination precision and efficiency of pollination (for example, see lines 68-69 where the authors seem to use efficiency and precision interchangeably) and cite key relevant studies that addressed some of the omissions highlighted above.

Finally, I noticed a few mistakes in References. For example, the last author is missing from Potente et al., 2022. (line 682).

Also, I noticed that "Polymorphic genera" is misspelled as "Polimorphic genera" at the top of Figure 2B.

REVIEWER COMMENTS

Reviewer #1 (Remarks to the Author):

General comments

The submitted manuscript by Simón-Porcar et al. investigates the macroevolutionary patterns of heterostyly, specifically genera that polymorphic for style length, across the angiosperms. The authors performed an extensive literature survey, multiple times, to find new cases of heterostyly, either documented with data or mentioned in studies. The known cases were expanded by about 1/5th. Different phylogenetic comparative methods were performed and multiple gains and losses were observed, more than previously shown/suggested. Analyses also showed gains associated with other floral traits. Overall, I first want to applaud the authors for the level of detail in searching the literature to make sure that all known cases were included in the study. With such a large-scale investigation, obtaining the necessary amount of data is no trivial task.

Response: We thank Reviewer #1 for this positive comment, indeed we think that this update was necessary and we are proud on the effort performed.

I personally think the Smith and Brown 2018 phylogeny is the best large phylogeny to use for most inferences. However, with only 85% of the polymorphic genera included, I have to ask if using the Janssens et al 2020 “A large-scale species level dated angiosperm phylogeny for evolutionary and ecological analyses” would be any better in terms of inclusiveness. I am not aware of as many taxonomic issues in the Smith and Brown phylogeny as was in the Zanne et al phylogeny that preceded it in large-scale analyses, but could a lack of taxonomic certainty in revised clades cause a drop in some retention of targeted genera?

We agree that Smith and Brown (2018) phylogeny is the best option for this type of inferences and, in fact it is still preferred over the phylogeny of Janssens et al (2020) (563 vs. 57 citations since 2020). The sampling of genera and species is much more comprehensive in the phylogeny of Smith and Brown (2018) than in the phylogeny of Janssens et al (2020). This is also true when considering only heterostylous genera and probably when considering only heterostylous species.

We agree that missing 15% of heterostylous genera in the phylogeny may have decreased the number of gains and losses of style-length polymorphism inferred. However, the genera sampled were evenly distributed along the phylogeny and therefore no bias is expected in our main conclusions. All the taxonomic incongruences arisen in heterostylous genera (i.e., synonyms) between the reports, the taxonomic treatment in the Plants of the World database and the GBOTB tree are shown in Supporting material 1. Some families are more prone to taxonomic changes than others and we prioritized the even distribution of genera over the sample size. Hence, to avoid a different treatment of heterostylous and non-heterostylous genera in the phylogeny and consequent possible biases in our phylogenetic sampling, we strictly followed the POWO treatment for all genera, even if a known synonym of a heterostylous genus appeared in the GBOTB tree.

Secondly, while I agree that providing a species level investigation is impossible, I am left wondering if the decision to prune to one tip per genus is artificially pushing more transition events to the roughly 4 million year mark? Based on the explained methodology, transitions in any given genus, regardless of size, age, and actual number of heterostyly transitions, would only be evident at the tip; regardless of the true evolutionary history of the trait or the possibility of multiple transitions occurring. For large, old genera I could see this really impacting the inferred dates. Using a large consensus tree such as the Smith and Brown tree would likely not be suitable for investigating the number of transitions within a genus due to low support values in some cases.

We understand the question of Reviewer #1 and agree on the fact that, by downsampling genera, terminal transitions that are rare within these genera (but sampled randomly) would be biased to be older than they would be inferred to be in a full species-level tree. Nevertheless we highlight that, in principle, Stochastic Mapping should sample transitions anywhere along terminal tips and, indeed, in this new version we entirely substituted the inference of gains and losses through Maximum Likelihood by Stochastic Mapping. As a result, we found that gains and losses were not clustered around the 4 million year mark, but they were more widely distributed across node ages in the angiosperms tree, as it is apparent in Figure 4b.

We also note that the random selection of one species per genus means a random representation of the age for all genera, regardless of whether they present style-length polymorphism or not. Hence, at the level explored, we would have not expected any major bias for the inferred ages of gains and losses and we don't think that this would be a critical issue for this study. Indeed, gains and losses can be various within genera, but this is impossible to assess at the level studied here and a number of studies within particular clades have attempted to do so.

Another major point I want to bring up is the difficulty in translating a "continuous" trait (stamen and carpel number) to a binary trait. My primary concern is if there is a biological threshold that makes a total of 10 stamens (and similar to five carpels) important. I understand the argument about lower number of floral parts, however is there any biological difference between 9 and 10 stamens or 10 and 11 stamens? Reconstructing stamen numbers as a "continuous" trait could be relevant, and having a better understanding the model of change (Brownian vs OU vs Early Burst) might tell something.

We understand the view of Reviewer #1 but we note that the thresholds chosen are somehow biologically meaningful. Ten is a common number of stamens for pentamerous flowers with two androecium whorls (or two perianth whorls), which are very common, particularly in Pentapetalae. Reductions (or variations) to trimerous and tetramerous are also common, but on the other hand increases in merism above five are less common. Hence, we believe that ten stamens and five carpels were appropriate thresholds to divide our dataset in taxa with a high or a low number of sex organs, considering the distribution of data for both traits. Importantly, we also repeated the analyses of all floral traits, except perianth symmetry, with different thresholds and obtained similar results.

Line 236: “We repeated the analyses on all floral traits (except perianth symmetry) using different thresholds for binary scoring and obtained similar results.”

We also note that understanding the evolution of the floral traits analysed was not the aim of our study, as it was to analyse their evolutionary correlation with the occurrence of style-length polymorphism. We highlight that BM and OU models are in a way even more simplistic/constrained than Mk models and, more importantly, may not be well suited to quantitative traits that do not vary continuously but instead as integers such as counts, as is the case for the number of sex organs. PGLS models analysing the correlation between a binary trait and a continuous one may also make potentially incorrect assumptions about continuous trait evolution. Given the nature of all traits analysed, we believe that Pagel’s models of correlated evolution seemed the most appropriate and straightforward approach.

Lastly, I do want to acknowledge the impressive amount of work that went into the manuscript, especially the data collection. However, after reading the manuscript I felt myself wanting more in terms of analyses and correlations/statements could really be made. While there is nothing wrong with the reconstruction methods, the title and summary suggest evidence for the association of heterostyly with specific animal pollination principles. This would be a fantastic thing to show, but on line 445 the authors even say that their results are independent of animal pollinators related to style-length polymorphisms. While the data is certainly useful on a large scale, and the authors suggest future studies that would be possible, I’m not sure the expectations of the title were met.

This comment is in line with the comments of Reviewer #2 and addressing it has entailed the major improvement of this new version of our manuscript.

We made an effort to collect data on the pollination system from virtually every study on style-length polymorphic species, and assembled a large dataset with the pollination system of a large number of angiosperm species included in the GBOTB tree to test the evolutionary association of style-length polymorphism with particular pollination systems. Specifically, we tested the association with biotic, insect, bird, generalist, and long-tongued pollination systems. We used Pagel’s models of correlated evolution based on species-level phylogenies because this is the taxonomic level at which several transitions between pollination systems take place. Interestingly, and despite our *a priori* expectations based on the frequencies of pollination systems in polymorphic species (see Supplementary Material S3), we found a significant association between the evolution of

style-length polymorphism and the long-tongued insect pollination system. Remarkably, our model found that transitions from style-length monomorphism to style-length polymorphism never occurred within short-tongued pollination systems. This result supports the literature that has regarded long-tongued insects (such as butterflies, long-tongued bees and long-tongued flies) as the main agents of the precise pollen transfer between morphs of heterostylous species.

Details on these new analyses are found in L242-280 (data search), L321-328 (data analyses) and L418-434 (results). See also Figure 6 and Supplementary material S3-S5.

Specific comments

Line 51-53: The last sentence of the summary talks about the association between trait evolution and animal pollination, yet by the end of the manuscript these two things are clearly not compared at all. While the hypothesis is attributed to Darwin, it does seem a little bit misleading since this cannot/was not addressed with the data on hand.

This comment has been responded above and addressed with new pollinator data sets.

Line 100: Seems a little bit odd to have just two citations that are 115 years apart. Are these the only two hypothesized examples?

Yes, these were indeed the authors proposing this hypothesis. Specifically, Lloyd and Webb rescued the original idea of Darwin, which had been largely disregarded in favour of genetic-based models such as the one of Charlesworth & Charlesworth (1979). L103.

Line 121: Unprecedented detail or unprecedented level/scope?

We think that “detail” is correct here, as other studies have assessed evolutionary patterns of heterostyly across all angiosperms but at a much higher taxonomic level (i.e. comparing plant orders; e.g., Naiki 2012). L128.

Line 158: There are several complex sentences here, so I’m a bit confused about what “it” is referring to here.

We have switched “this condition” and “it” within this sentence and think that it is now clearer that we are always referring to “style-length dimorphism”.

L168: “This definition included style-length dimorphism and we also included reports of this condition because Lloyd and Webb’s (1992b) model (see also Charlesworth & Charlesworth, 1979; Simón-Porcar, 2018) proposes it as an evolutionary precursor of heterostyly that promotes disassortative mating in a very similar manner (Cesaro & Thompson, 2004; Simón-Porcar et al., 2015a; 2022).”

Line 179: The website appears to be powo.science.kew.org

Thanks for noting this. We have corrected the web address. **L190**.

Line 244-246: Maybe I'm missing something but why was corHHM used in some cases, while ML (ASR?) and stochastic mapping were used in others? Also, no details were provided as to what programs were used for these later analyses. I'm guessing phytools.

We used Hidden Markov models (corHHM) to fit the best model for the evolution of style-length polymorphism in our phylogeny. In this new version, we excluded the ML analyses and used only stochastic mapping to map the gains and losses of style-length polymorphism based on the best corHHM evolutionary model.

L290: “We estimated the transition rates between states and ancestral state of style-length polymorphism (coded as a presence/absence binary trait) across the angiosperm phylogeny with Hidden Markov models (HMM) as implemented in the function corHMM of the R package *corHMM* (Beaulieu et al., 2017). HMM models allow for some level of heterogeneity across lineages while keeping the number of parameters low. We used these models because the distribution of style-length polymorphism across angiosperm phylogeny suggested high heterogeneity in the evolution of the trait. We ran the ‘*equal rates*’ (equivalent to the ‘*symmetric rates*’ in the case of binary traits) and ‘*all rates different*’ models with a single or two transition rate categories each (four models in total). We used the Akaike information criterion (AIC) to select the best fitting model, which was used to reconstruct the evolution of style-length polymorphism in the phylogeny through stochastic character mapping. We ran 100 simulations to estimate the number and age of gains and losses of style-length polymorphism across angiosperm genera using the makeSimmap function in corHMM.”

We also clarify why we used corHHM for these analyses, and phytools for Pagel models. We used fitPagel instead of corHMM because they are homologues when using a single transition rate category and the former is much more friendly for custom matrices for correlation than the latter. We ran initial trials of corHMM models with two transition rate categories, but these included too many parameters and displayed a low fit to our data, so we discarded using two rate categories.

L310: “We used Pagel’s models (Pagel, 1994) for two binary traits as implemented in the function fitPagel in the R package *phytools* 1.5.1 (Revell, 2012). We used fitPagel instead of corHMM because (i) they are equivalent when using a single transition rate category, and (ii) corHMM models with two transition rate categories included too many parameters and initial trials displayed a low fit to our data.”

Line 249-250: Did the age estimates include the uncertainty in divergence times from the Smith and Brown tree, or what is just a point estimate for the consecutive nodes?

We note that the GBOTB tree have fixed ages and hence no uncertainty around node ages.

In the previous version, the age estimates for each transition were based on ML analyses and were just the midpoint estimate for the consecutive nodes, so they did not have uncertainty either. In our new version we estimated the number of transitions (including their location and ages) based on 100 simulations of stochastic mapping which gives us a measure of uncertainty in our transition estimates.

Figure 3: “Transition rates between R1 and R2, and polymorphic and monomorphic states, are shown in bold in (C), jointly with the number of transitions and their confidence intervals based on 100 stochastic mapping simulations.”

Line 256: Truly random? If more than three, why not take the majority consensus?

In an ultrametric phylogeny (all tips end in the present) there are not differences if you choose one species or other (or a consensus).

Line 258: version of *phytools*?

We noted that we used *phytools* 1.5.1. **L311**.

Line 307: Is this maximum probability value appropriate since the events are not entirely independent?

The gains and losses inferred across the phylogeny are independent and hence we consider it informative to build the probability density plot (**Figure 4b**) and to provide these values. We have also added a figure on the accumulated number of lineages within each transition rate (R1 and R2) and style-length polymorphic state (polymorphism, monomorphism) along evolutionary time (**Figure 4a**).

Line 312: This partially answers a previous question in regards to confidence intervals. How much confidence can we have then in these point estimates?

This comment has been responded above. We still consider these values as informative.

L369: “The most ancient and most recent gains, respectively, were dated at 86.01 and 0.02 Myr ago, with a maximum probability at 6.6 Myr (mode estimated with the multimode R package; Ameijeiras-Alonso et al., 2018). The most ancient and most recent losses, respectively, were dated 68.37 and 0.22 Myr ago, with a maximum probability at 6.8 Myr. The ages given above correspond to the ages estimated from a single simulation in stochastic mapping and do not include the inherent uncertainty of divergence time analyses.”

Line 313 (Figure 3): If the tree in one panel is going to have clade arc labels, it would be helpful if both had major lineages/important lineages marked.

We see it as appropriate to label major angiosperm lineages in this figure and we did so

in **Figure 3a**. We did not label them again in panel B as it looked too overloaded and seemed repetitive.

Line 385: Agreed, especially if within a genus there were multiple independent transitions, which large genera may likely be the case.

We agree and as stated above this needs to be analysed for those particular genera.

Line 407: I know acctran and deltran are parsimony based, but could this observation be due to a bias in the inference method, somehow tied to the speed of evolution? Or is it due to the pruning method? If more than one species per genus was kept, its reasonable for changes involving multiple species to occur not at tips but a few nodes back.

This comment referred to a result that has changed after replacing ML by stochastic mapping analyses. Indeed, we have now found that gains and losses were distributed along most the evolutionary time of angiosperms (**Figure 4b**).

L367: “From this model, we inferred 152 independent gains of style-length polymorphism and 137 independent losses that generally appeared during most of angiosperms evolutionary time (Fig. 4b).”

Line 415-416: This is a bit confusing since this sentence suggest gains and losses are almost equal, but then the next sentence is 102:25 ratio.

These results have changed with stochastic mapping, and we have made an effort to explain them clearly. We must take into account the prevalence of the ancestral monomorphic state when comparing transition rates and the number of gains and losses retrieved in our analyses. Indeed, transition rates and actual transitions are different measures that may even present opposite asymmetries. Details are found in **L356-374**, and in **Figure 3** and **Figure 4**.

Line 418: I'm not sure if I buy this argument entirely. Without a reliable fossil representation, this seems a bit oversimplified given any transition in a genus would only be evident at the tip, regardless of the size and age of a given genus and where in its evolutionary history one or multiple transitions occurred.

This argument has been removed after the new results retrieved with stochastic mapping.

Line 420-422: While fundamentally I agree with this sentence, providing some citations or specific reasoning would be useful.

Following a comment of Reviewer #2, we extended the discussion on the potential role of style-length polymorphism on diversification rates, including meaningful references in the following sentences. We highlight that most evidence to date suggest the context or lineage dependent role of traits such as heterostyly on diversification rates.

L508: “Whether style-length polymorphism was associated with different speciation rates or extinction rates in R1 and R2 cannot be known with our current knowledge of phylogenetics and distribution of style-length polymorphism across all angiosperm species (Sauquet & Magallón, 2018). At the genus level there is mixed phylogenetic evidence about the role of heterostyly in diversification rates (de Vos et al., 2014b; Maguilla et al., 2021; Cohen, 2022; see also Haller et al., 2014), which is congruent with the apparent context and/or lineage-dependent role of floral traits on angiosperms diversification rates (Helmstetter et al., 2022).”

Line 446: knowing the specific selective pressures is difficult. If it is floral architecture, there is likely some selective forces that are driving the change. Is it outside the realm of feasibility to look at style-length polymorphisms vs selfing/outcrossing or more importantly animal pollination versus other vectors?

As exposed above, we added new analyses on the correlation of style-length polymorphism with pollination systems. Addressing the correlation of selfing vs. outcrossing seems beyond the scope of this article and in addition we are not sure if data are available in the literature as to cover enough taxa. Remarkably, selfing/outcrossing rates are also highly variable also across populations and years (see Whitehead et al. 2018, *Front. Ecol. Evol.*).

Line 454-455: This partly addresses my previous question, even if this sort of question/association is highly interesting.

This comment has been responded to above.

Reviewer #2 (Remarks to the Author):

The paper provides new, valuable results on the distribution of heterostyly in angiosperms as well as the number and timing of origins and losses of the trait. The paper is well written and clear. The analyses appear to be thorough and well executed. In general, the paper represents a valuable contribution to studies of floral heteromorphism.

However, I wonder whether the main conclusion of the paper, namely, that heterostyly evolved in response to selection for increased pollination precision, is warranted. I will try to unpack the sources of my concern.

We thank Reviewer #2 for the positive comments on the former version of our manuscript. Indeed, the concern exposed was partly shared with Reviewer #1. In this new version, we made an effort to clarify the meaning of the pollination-precision hypothesis tested and included new analyses on the association of style-length polymorphism with pollination systems. We provide detailed responses on every comment below.

On the concept of “precision of pollination”. The authors state that Darwin proposed the hypothesis of increased pollination precision for the origin of heterostyly. However, did

Darwin really focus on “pollination precision” or, rather, disassortative pollination and promotion of outcrossing, which decrease the harmful effects of inbreeding? The hypothesis of disassortative pollination (hence promotion of outcrossing) is a quantitative model requiring that more pollen grains are transferred to reciprocal and compatible stigmas than to non-reciprocal and incompatible stigmas.

I think that a general problem affecting the main conclusion of the paper is that the authors fail to define what they mean with “pollination precision”, and this omission needs to be corrected. Importantly, the authors should link their hypothesis with other studies that have addressed the role of pollination precision in angiosperm evolution and the importance of precision of sexual organ reciprocity in heterostylous taxa (for example, Armbruster et al., *New Phytologist*, 2022 and 2017, respectively).

We agree with Reviewer #2 and think that this concern was raised because of the need to better define “pollination precision” and the relationship between this hypothesis and the Darwinian hypothesis about the function and evolution of heterostyly. As Stewart et al. (2022), on which the commentary of Armbruster (2022) is based, we define “pollination precision” as the “*increased pollination efficiency achieved by enhancing pollen deposition on precise areas of the pollinator*”.

L72: “According to the *pollination-precision hypothesis*, reproductive efficiency is increased by floral traits improving the fit of pollinators to flowers and their reliable contact with sex organs, thus improving pollen deposition on precise areas of pollinators’ bodies (e.g. Pauw, 2006; Armbruster 2022; Stewart et al., 2022).”

We highlight that, on the basis of the Darwinian hypothesis for heterostylous plants, disassortative mating (and hence the promotion of outcrossing) rely on disassortative pollination, which depends, in turn, on the precise placement of pollen on the pollinator’s body. Hence, we consider that, indeed, the foundation of the Darwinian hypothesis is the pollination-precision hypothesis, and this is what we are testing in this article.

L42: “Based on his archetypal heterostylous flower, including actinomorphy, few stamens and a tube, Darwin hypothesized that heterostyly evolved to promote outcrossing through efficient disassortative pollen transfer involving different areas of a pollinator’s body, thus proposing his seminal *pollination-precision hypothesis*.”

L103: “Based on this floral archetype, Darwin (1877b) and Lloyd and Webb (1992b) hypothesized that heterostyly evolved to promote cross-fertilization through disassortative mating (as in dioecious plants but maintaining both male and female functions in the same flower) through efficient pollen transfer between morphs that involves different areas of a pollinator’s body (Fig. 1). Hence, the basis of the Darwinian hypothesis about the evolutionary meaning of heterostyly was the *pollination-precision hypothesis*.”

At the macroevolutionary level considered in our study, we test the pollination-precision hypothesis by testing the evolutionary correlation of style-length polymorphisms with floral traits and pollination systems favouring the precise contact of pollinators bodies with floral sex organs, i.e., enhancing pollen deposition on precise areas of the pollinator.

L122: “Macroevolutionary testing for correlation of heterostyly with floral traits and pollination systems that enhance pollen deposition on precise areas of pollinators’ bodies is an idea that was suggested by Lloyd and Webb (1992a) at a time when the analytical tools required were scarcely developed.”

We thank Reviewer #2 for the suggested references and, indeed, we find the first one to be highly relevant for our study. We have included this reference, jointly with that of Steward et al. (2022), in a more explicit explanation of the “pollination precision” concept and our hypothesis in the Introduction (see citation of **L72** above).

At the macroevolutionary level considered here, in which we could not even distinguish between heterostylous and other style-length dimorphic conditions, we felt that the concept of reciprocity falls beyond the scope of our study and we think that to explicitly discuss it may be misleading. It is impossible to assess sexual reciprocity, a population-based feature, at the macroevolutionary level studied here. Nevertheless, we think that the reference of Armbruster et al. (2017) is a highly valuable contribution to the study of pollination accuracy in heterostylous plants at the microevolutionary level and we included it where we considered appropriate.

L543: “Future efforts assessing the macro-evolutionary correlation of style-length polymorphism with pollinator types, including the assessment of their behaviour and precise pollen deposition areas (e.g. Stewart et al., 2022), their efficiency in promoting disassortative pollination (as done for some specific cases; e.g. Stone & Thomson, 1994; Lau & Bosque, 2003; Simón-Porcar et al., 2014; Armbruster et al., 2017), and their interplay with floral traits, would seem the most promising additional test of the *pollination-precision hypothesis* in this context.”

In different parts of the paper the authors refer to “floral tubes promoting the precise contact between pollinators and floral sexual organs” (line 438), but again they do not define what they mean with “precise contact”. If they mean to say that pollen is deposited on discrete, non-overlapping parts of the pollinators’ body, as represented in their Fig. 1, this is not the case, or, rather, it is not as simple as that (see quantitative definition of disassortative pollination above). Some of the few studies that quantified disassortative pollination via pollen counts found that, while the hypothesis of disassortative pollination generally holds, pollen grains are far from being deposited in discrete, non-overlapping parts of the pollinators’ bodies. Indeed, pollen grains from high and low anthers are found in the same parts of the pollinators’ bodies, but in different proportions (see Keller et al. 2014, *Functional Ecology*).

We define precise contact as a reliable constant contact of pollinators bodies with floral sex organs that enhances pollen deposition on precise areas of the pollinator (see citation of **L72** above).

We agree with this comment of Reviewer #2, but we note that this mechanism, regardless of whether it functions perfectly or imperfectly in particular taxa or populations, is ultimately dependent upon a floral architecture and pollination system that supports

precise pollination in the way defined here. At the macroevolutionary level studied here, precise pollination cannot mean that pollen is always located perfectly in differentiated discrete parts of the pollinator body. Testing such discrete pollen segregation is only possible with empirical data in populations and species, which has hardly ever been attempted and which is certainly not feasible in our comparative angiosperm-wide analysis (but see citation of L543 above). However, the issue here is that different floral traits may favour more precise pollen deposition and that it may happen with particular pollinator types, and our correlative approach supports that.

L109: “Additionally, Ganders (1979) suggested that the floral tube promotes precise contact between the plants’ sex organs and the pollinators’ body, and there is extensive literature (e.g. Darwin, 1877b; Beach & Bawa 1980; Lloyd & Webb 1992b; Stone 1996; Simón-Porcar et al., 2014) regarding long-tongued pollinators probing for nectar as being agents of precise pollination in heterostylous flowers.”

L139: “For the first time, we demonstrate that style-length polymorphism originated repeatedly and independently across genera in lineages with flowers and pollination systems favouring precise pollen transfer.”

See L210-240 for detailed descriptions of how the floral traits can improve pollination precision.

L276: “We tested the hypothesis that style-length polymorphism evolved in association with long-tongued insect or bird pollination systems, as these animals, probing for floral nectar, have been regarded as the main agents promoting precise pollination (Darwin, 1877b; Beach & Bawa 1980; Lloyd & Webb 1992b; Stone 1996; Simón-Porcar et al., 2014).”

Additionally, I wonder how one can reconcile the concept of “pollination precision” as the major driver for the evolution of heterostyly (rather than disassortative pollination and outcrossing) with the fact that, as the authors acknowledge (line 453-454), heterostylous taxa are typically pollinated by multiple pollinators and that pollen deposition on their bodies does not occur in discrete pollen packages (as for example, pollinia), but rather in a rather messy “pollen cloud” (as discussed above).

See responses above. In fact, although we found that most heterostylous species studied are pollinated by wide arrays of pollinators (see Supplementary Material S3), our new analyses revealed that style-length polymorphism was associated with long-tongued insects at the macroevolutionary scale. Pollination by short-tongued insects was associated with the loss of style-length polymorphism. This might be one of the reasons why style-length polymorphism is lost so frequently.

L420: “The model of dependent evolution with style-length polymorphism presented greater support than the corresponding model of independent evolution only for long-tongued vs. short-tongued insects pollination systems (Supplementary Material S4). The transition rates between states in this model supported the highest probability of the style-length polymorphic state with long-tongued insect pollination (Fig. 6). The missing

transition from style-length monomorphism to style-length polymorphism in short-tongued pollination systems is especially remarkable (Fig. 6).”

Moreover, if pollination precision were the key driver for the repeated origin of heterostyly, it would imply that heterostyly might also spur an increase of speciation rates via a form of mechanical reproductive isolation linked with the “precise” partitioning of flower-pollinator space for the uptake and deposition of pollen grains. However, previous studies found complex results on whether the precision of pollen transfer can drive divergence. A modeling study concluded that precision of pollen transfer can have a small effect on population divergence, but this effect is very context-dependent (Haller et al., 2014, PLOS One). A macroevolutionary study found that heterostyly is linked with an increase of diversification rates, but this effect does not depend on an increase of speciation rates, rather on a decrease of extinction rates, possibly because heterostylous taxa are more resilient to changes of environmental conditions because they harbor more genetic diversity than non-heterostylous taxa (deVos et al., 2014. Proc. R. Soc. B).

We appreciate the comment of Reviewer #2 and we have extended our discussion to highlight that most evidence to date suggest the context or lineage dependent role of traits such as heterostyly on diversification rates. Our former version already included the citation of deVos et al (2014) and we have now added the citation of Haller et al (2014).

L508: “Whether style-length polymorphism was associated with different speciation rates or extinction rates in R1 and R2 cannot be known with our current knowledge of phylogenetics and distribution of style-length polymorphism across all angiosperm species (Sauquet & Magallón, 2018). At the genus level there is mixed phylogenetic evidence about the role of heterostyly in diversification rates (de Vos et al., 2014b; Maguilla et al., 2021; Cohen, 2022; see also Haller et al., 2014), which is congruent with the apparent context and/or lineage-dependent role of floral traits on angiosperms diversification rates (Helmstetter et al., 2022).”

We understand the comment of Reviewer #2 in the context on his/her view of the pollination-precision hypothesis in our previous submission. However, we note that the hypothesis posed and the two cited studies discuss the role of heterostyly on diversification rates in the context of changes in reciprocity and consequent reproductive isolation of populations. However, our study was conducted at the genus level and, hence, we think that our results cannot be directly related to this hypothesis. As noted above, our analyses and results are independent of population-level features such as reciprocity and reproductive isolation. We believe that the association of style-length polymorphism with particular floral traits and pollination systems across all angiosperms support the pollination-precision hypothesis, but this would not necessarily imply that style-length polymorphism increases diversification rates across all angiosperms through the precise pollen transfer.

The main thesis of the article is that heterostyly evolved in response to selection for increased pollination precision (rather than to promote disassortative pollination and prevent selfing, with the combined effect of increased outcrossing). If indeed pollination precision is the driver for the evolution of heterostyly, then the authors should also discuss

the specific selective advantages for heterostylous species (as compared to non-heterostylous species) of having increased “precision” vs. the advantages of elevated outcrossing, which does not directly depend on “precision” per se.

We highlight that the promotion of outcrossing through disassortative mating is a hypothesis dependent on the pollination-precision hypothesis as defined here. Outcrossing and disassortative mating are a consequence of pollination-precision, which we can consider the underlying mechanism.

To conclude, I think the authors need to provide a precise definition of what they mean with “pollination precision” and how they distinguish its selective advantages from those of disassortative pollination and outcrossing. They should also distinguish the concepts of pollination precision and efficiency of pollination (for example, see lines 68-69 where the authors seem to use efficiency and precision interchangeably) and cite key relevant studies that addressed some of the omissions highlighted above.

See comments above. We consider pollination efficiency as a product of pollination precision. We emphasize that as proposed by Darwin, pollination precision is a prerequisite for disassortative pollination. We have rewritten the relevant parts of the manuscript to clarify the ideas exposed here. Unfortunately, we are not aware of any studies directly relating pollination precision (i.e. segregation of pollen in pollinator bodies) and pollination efficiency in heterostylous species.

L42: “Based on his archetypal heterostylous flower, including actinomorphy, few stamens and a tube, Darwin hypothesized that heterostyly evolved to promote outcrossing through efficient disassortative pollen transfer involving different areas of a pollinator’s body, thus proposing his seminal pollination-precision hypothesis.”

L72: “According to the *pollination-precision hypothesis*, reproductive efficiency is increased by floral traits improving the fit of pollinators to flowers and their reliable contact with sex organs, thus improving pollen deposition on precise areas of pollinators’ bodies (e.g. Pauw, 2006; Armbruster 2022; Stewart et al., 2022).”

Figure 1: “Graphic representation of heterostyly. Graphic representation of the two floral morphs (L=long-styled morph, S=short-styled morph) of a distylous species and the hypothetical mechanism of pollen transfer between morphs in differentiated parts of a pollinator’s body, based on the pollination-precision hypothesis.”

L103: “Based on this floral archetype, Darwin (1877b) and Lloyd and Webb (1992b) hypothesized that heterostyly evolved to promote cross-fertilization through disassortative mating (as in dioecious plants but maintaining both male and female functions in the same flower) through efficient pollen transfer between morphs that involves different areas of a pollinator’s body (Fig. 1). Hence, the basis of the Darwinian hypothesis about the evolutionary meaning of heterostyly was the *pollination-precision hypothesis*. Additionally, Ganders (1979) suggested that the floral tube promotes precise contact between the plants’ sex organs and the pollinators’ body, and there is extensive literature (e.g. Darwin, 1877b; Beach & Bawa 1980; Lloyd & Webb 1992b; Stone 1996; Simón-Porcar et al., 2014) regarding long-tongued pollinators probing for nectar as being agents of precise pollination in heterostylous flowers.”

L133: “We demonstrate that style-length polymorphism originated repeatedly and independently across genera in lineages with flowers and pollination systems favouring precise pollen transfer. Our results provide strong support for the pollination-precision hypothesis as a basis of the paradigmatic convergent evolution of heterostyly across angiosperms.”

Finally, I noticed a few mistakes in References. For example, the last author is missing from Potente et al., 2022. (line 682).

We thank Reviewer #2 for noticing this mistake, we fixed it and reviewed the entire References section to ensure that it does not contain further mistakes.

Also, I noticed that “Polymorphic genera” is misspelled as “Polimorphic genera” at the top of Figure 2B.

We thank Reviewer #2 for noticing this misspelling, which has been fixed.

Reviewers' Comments:

Reviewer #1:

Remarks to the Author:

I appreciate the thoroughness that the authors addressed the comments/concerns raised in the previous round of review. After reading thru the response letter and the manuscript, I feel like all of my concerns were more than adequately addressed by the authors. The additional analyses and revisions make the manuscript much stronger than the original version. I'm more than happy to recommend this revised version for publication and think it will be a valuable contribution.

Reviewer #2:

Remarks to the Author:

The authors addressed the concerns raised by the reviewers. Notably, they generated new pollinator data sets and performed new analyses in response to a similar comment by both reviewers that challenged whether the authors could defend their claim (also asserted in the title of the submitted paper) that they found an association between heterostyly and specific animal pollination systems.

While some disagreements about different interpretations of proximate vs. ultimate mechanisms driving the evolution of style polymorphism may remain, I believe the submitted paper contributes to advancing the discourse about the evolution of this trait across angiosperms.

Some of the contrasting interpretations of the key mechanisms driving the evolution of stilar polymorphism (i.e., precision vs. disassortative pollination) may stem, at least in part, from the different perspectives provided by macro- vs. micro-evolutionary approaches. Whether such contrasting views can be readily reconciled or not represents a classic challenge running throughout evolutionary biology. Ultimately, both perspectives are needed to advance our knowledge on the patterns and mechanisms of trait evolution. Thus, I think this paper will make a valuable contribution to the field of stilar polymorphism.

Response to Reviewers

Convergent evolutionary patterns of heterostyly across angiosperms support the pollination-precision hypothesis'' (NCOMMS-23-06821A)

REVIEWERS' COMMENTS

Reviewer #1 (Remarks to the Author):

I appreciate the thoroughness that the authors addressed the comments/concerns raised in the previous round of review. After reading thru the response letter and the manuscript, I feel like all of my concerns were more than adequately addressed by the authors. The additional analyses and revisions make the manuscript much stronger than the original version. I'm more than happy to recommend this revised version for publication and think it will be a valuable contribution.

Response: On behalf of all authors, I thank Reviewer #1 for his/her comments and suggestions on the previous version of our manuscript, which significantly contributed to its improvement. We are grateful as well for his/her very positive view on this revised version.

Reviewer #2 (Remarks to the Author):

The authors addressed the concerns raised by the reviewers. Notably, they generated new pollinator data sets and performed new analyses in response to a similar comment by both reviewers that challenged whether the authors could defend their claim (also asserted in the title of the submitted paper) that they found an association between heterostyly and specific animal pollination systems.

While some disagreements about different interpretations of proximate vs. ultimate mechanisms driving the evolution of style polymorphism may remain, I believe the submitted paper contributes to advancing the discourse about the evolution of this trait across angiosperms.

Some of the contrasting interpretations of the key mechanisms driving the evolution of stylar polymorphism (i.e., precision vs. disassortative pollination) may stem, at least in

part, from the different perspectives provided by macro- vs. micro-evolutionary approaches. Whether such contrasting views can be readily reconciled or not represents a classic challenge running throughout evolutionary biology. Ultimately, both perspectives are needed to advance our knowledge on the patterns and mechanisms of trait evolution. Thus, I think this paper will make a valuable contribution to the field of stylar polymorphism.

Response: On behalf of all authors, I thank Reviewer #2 for his/her comments and contribution to improve our manuscript. We totally agree on the need and complementary role of studies at micro and macroevolutionary scales to increase our understanding of the evolution and function of heterostyly, and of any other variation in biological traits.